# Macroalgae

## Leonel Pereira

Marine and Environmental Sciences Centre (MARE), Department of Life Sciences, University of Coimbra, 3000-456 Coimbra, Portugal; leonel.pereira@uc.pt

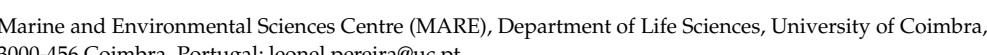

**Definition:** What are algae? Algae are organisms that perform photosynthesis; that is, they absorb carbon dioxide and release oxygen (therefore they have chlorophyll, a group of green pigments used by photosynthetic organisms that convert sunlight into energy via photosynthesis) and live in water or in humid places. Algae have great variability and are divided into microalgae, small in size and only visible through a microscope, and macroalgae, which are larger in size, up to more than 50 m (the maximum recorded was 65 m), and have a greater diversity in the oceans. Thus, the term "algae" is commonly used to refer to "marine macroalgae or seaweeds". It is estimated that 1800 different brown macroalgae, 6200 red macroalgae, and 1800 green macroalgae are found in the marine environment. Although the red algae are more diverse, the brown ones are the largest.

**Keywords:** macroalgae; classification; pigments; morphological characteristics; reproduction

## 1. Introduction

Algae are single or multicellular organisms that live in water or in humid places. These organisms have chlorophyll (an organic pigment capable of absorbing and channeling the energy of sunlight), which is why they are able to perform photosynthesis, that is, the transformation of luminous energy into chemical energy, capturing carbon dioxide ($CO_2$) to form complex organic compounds (along with water and mineral salts), and releasing gaseous oxygen ($O_2$), during the process of organic synthesis. Algae are considered the true "lungs" of planet Earth, stealing this epithet from large forest patches, such as the Amazon rainforest. Algae are distinguished from seagrass (angiosperms) because, unlike the latter, they do not have a vascular system (xylem and phloem). Algae on the seafloor have a holdfast and transport nutrients through the body by diffusion, while seagrasses are flowering vascular plants with roots and an internal transport system [1,2].

## 2. Microalgae versus Macroalgae

Algae are, therefore, an autotrophic group (capable of producing their own food through an anabolic process) and one where there is a great diversity of organisms, regarding morphology, the degree of complexity of their body structure, and size. Due to this great variability, algae are generally divided into microalgae (Figure 1a), only visible through magnification equipment, and macroalgae, perfectly visible to the naked eye (Figure 1b).

Microalgae constitute the phytoplankton and thus live in surface waters, suspended in the water column and restricted to the euphotic zone. Phytoplankton constitute the base of the marine food chain-feeding invertebrates, such as crustaceans (e.g., copepods, which are part of the zooplankton), mollusks (e.g., mussels), or vertebrates (such as juvenile sardines, among other species).

Macroalgae or seaweeds are macroscopic marine algae that can reach several meters in length (some thalli of these algae can reach 65 m in length, if not subject to intense herbivory). As primary producers, macroalgae are at the base of the marine food chain supporting several communities of herbivorous animals (invertebrates, such as some sea urchins and/or gastropods, and vertebrates, such as herbivorous fish) where they end

up finding refuge from their predators, the carnivores. In order to escape this herbivory, which is sometimes intense (in natural reefs or rocky walls of the continental shelf), many macroalgae have been improving defense strategies, with calcification (common in red algae, such as the Corallinaceae family) being one of the most common, although they can also produce secondary metabolites (terpenes, aromatic substances, polyphenols, etc.) that act as demotivators of their ingestion [3].

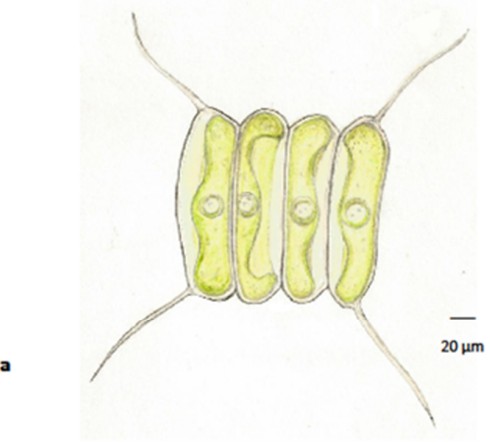

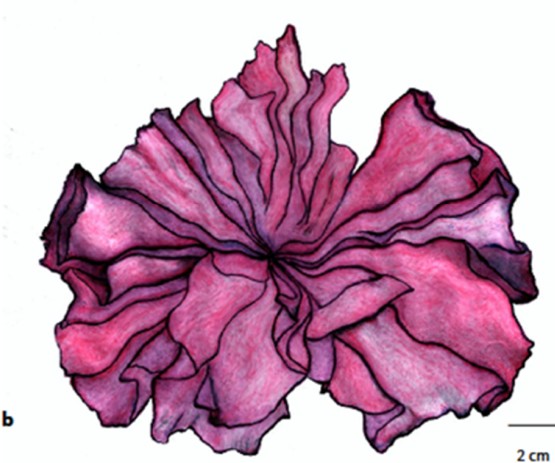

**Figure 1.** Algae: (**a**) microalgae are small algae, requiring the use of a microscope for observation (*Scenedesmus*, Chlorophyta); (**b**) macroalgae are larger algae, visible to the naked eye, in general marine and sometimes with considerable dimensions (reaching more than 60 m in length), whose thalli can have a high degree of complexity (*Porphyra umbilicalis*, Rhodophyta).

## 3. Ecology

Two environmental requirements dominate the ecology of seaweed. They live in seawater (or at least brackish water) and need enough light for photosynthesis to occur. Another common requirement is a fixation point, and therefore marine algae most commonly inhabit the coastal zone (waters close to the coast) and within that zone, on rocky sea fronts, rather than on sand or gravel. In addition, there are few genera (for example, *Sargassum*) that do not live attached to a rocky substrate, but float freely [3]. Seaweed occupies several ecological niches. On the surface, they are moistened only by sea foam, while some species can stick to a substrate several meters deep. In some areas, coastal seaweed colonies can extend for miles to the sea. The deepest living algae are some species of red algae. Others have adapted to live in tide pools. In this habitat, seaweed must resist rapid changes in temperature and salinity and occasional drying, at the pace of alternating tides [1,3].

Macroalgae and macroalgae debris has also been shown to be an important source of food for benthic organisms. These macroalgae fronds tend to be used by benthic animals in the intertidal zone near the coast. Alternatively, pneumatocysts ("bubbles" filled with gas) can keep the macroalgae thalli afloat, and the fronds are transported by wind and currents from the coast to the deep ocean [2,3].

As macroalgae absorb carbon dioxide and release oxygen in photosynthesis, the fronds of macroalgae can also contribute to carbon sequestration in the ocean, through which the leaves of macroalgae drift from the coast to the deep ocean basins and sink to the bottom of the sea without being remineralized by the organisms. The importance of this process for the storage of blue carbon is currently studied and discussed among scientists [3,4].

## 4. Pigment Composition and Classification

The coloring of an alga is nothing more than the visible expression of the combination of the different photosynthetic pigments present in its cells. For this reason, it has been more than a century since the distinction of the different phyla and classes of marine macroalgae was made with the help of their coloring. Macroalgae have extremely varied colors, but all of them have chlorophyll. This pigment is inside small organelles, the chloroplasts, responsible for the green coloring of many plants (vascular and non-vascular).

Macroalgae are thus aquatic photosynthetic organisms (mainly marine) belonging to the domain Eukarya and the kingdoms Plantae (green and red algae) and Chromista (brown algae). Although classification systems have evolved a lot over time, it is generally accepted that (Figure 2, Table 1) [3]:

1.  Green macroalgae are included in the phylum Chlorophyta, and their pigmentation is identical to that of vascular plants (chlorophylls a and b and carotenoids);
2.  Red macroalgae belong to the phylum Rhodophyta; they have chlorophyll a, phycobilins, and some carotenoids as photosynthetic pigments;
3.  Brown macroalgae belong to the phylum Ochrophyta, and all of them are grouped in the class Phaeophyceae; their pigments are chlorophylls a and c and carotenoids (where fucoxanthin predominates, responsible for their brownish color);
4.  Blue-green algae are included in the phylum Cyanobacteria; they have chlorophyll a (green), carotenoids (yellow), phycocyanin (blue), and, in some species, phycoerythrin (red) (both phycobilins).

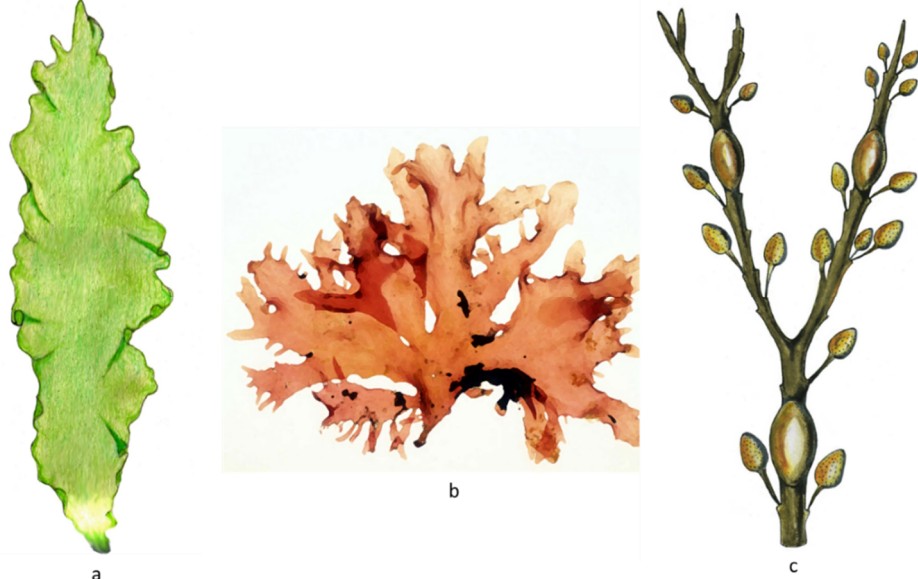

**Figure 2.** The three main taxonomic groups of macroalgae: (**a**) phylum Chlorophyta—green algae; (**b**) phylum Rhodophyta—red algae; (**c**) phylum Ochrophyta, class Phaeophyceae—brown algae.

**Table 1.** Classification and pigment composition characteristic of some algae [1].

| Phylum (*) | Class | Chlorophylls | Carotenoids | Phycobilins |
|---|---|---|---|---|
| Cyanobacteria | Cyanophyceae (**) | a | β-carotene, myxoxanthophyll, zeaxanthin | C-phycocyanin (+) C-phycoerythrin (−) |
| Chlorophyta | Bryopsidophyceae Siphonocladophyceae Ulvophyceae | a, b | β-carotene, lutein, neoxanthin, violaxanthin, zeaxanthin | - |
| Rhodophyta | Bangiophyceae Florideophyceae | a, d | β-carotene, lutein, zeaxanthin | R-phycocyanin (−) R-phycoerythrin (+) |
| Ochrophyta | Phaeophyceae | a, c | β-carotene, fucoxanthin, violaxanthin | |

(*) In addition to these phyla, there are others that are not mentioned here because they include only microscopic algae [1]. (**) In general, species of the class Cyanophyceae, phylum Cyanobacteria, are not easy to find on the coast; however, the species Rivularia bullata may be present on the marine coasts. (+) Present in greater %, and (−) present in minor %.

## 5. Morphological Types

Thallus (plural "thalli") is the plant body of algae, fungi, lichens, and some liverworts, a plant body that is not differentiated into stem and leaves and lacks true roots and a vascular system.

The thalli of most macroalgal species are erect, especially when immersed, unlike some species whose thalli are prostrate, formed by thin discs or by incrustations, adherent to the substrate. The thallus of a macroalgae is divided into the "frond", the upright part, which presents a great variability in shape and size, and a fixation organ. These variations can be accentuated by the external environmental conditions where they develop and can be very evident in populations of the same species, as in *Chondrus crispus* and *C. crispus* var. *filiformis* (Figure 3), which are common in coastal regions with high wave exposure in North Atlantic European shores.

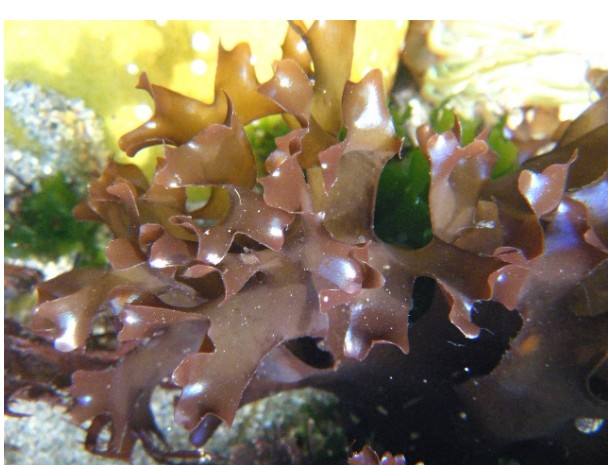 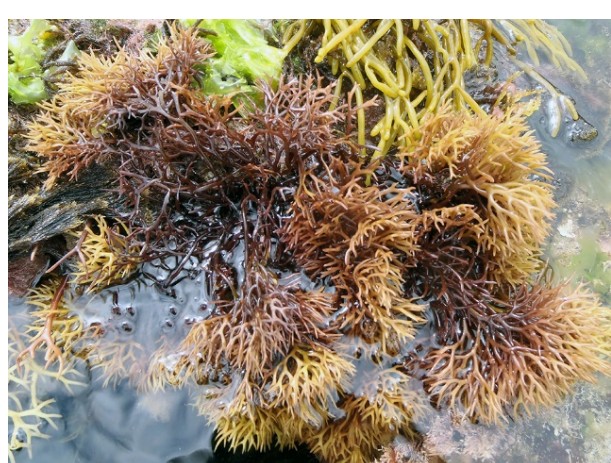

**Figure 3.** Red macroalgae *Chondrus crispus* (**left**) and *Chondrus crispus* var. *filiformis* (**right**).

### 5.1. Main Morphological Characteristics of Macroalgae

The shape or morphology of a given thallus (macroalgal body; plural: thalli) is a very useful feature to distinguish the various species of macroalgae. In addition, different species present different consistencies or textures to the touch, which can help us distinguish them [2].

Only the large macroalgae (kelp) present a more robust fixation part, composed of more or less curved elements, the hapteron or holdfast (Figure 4e).

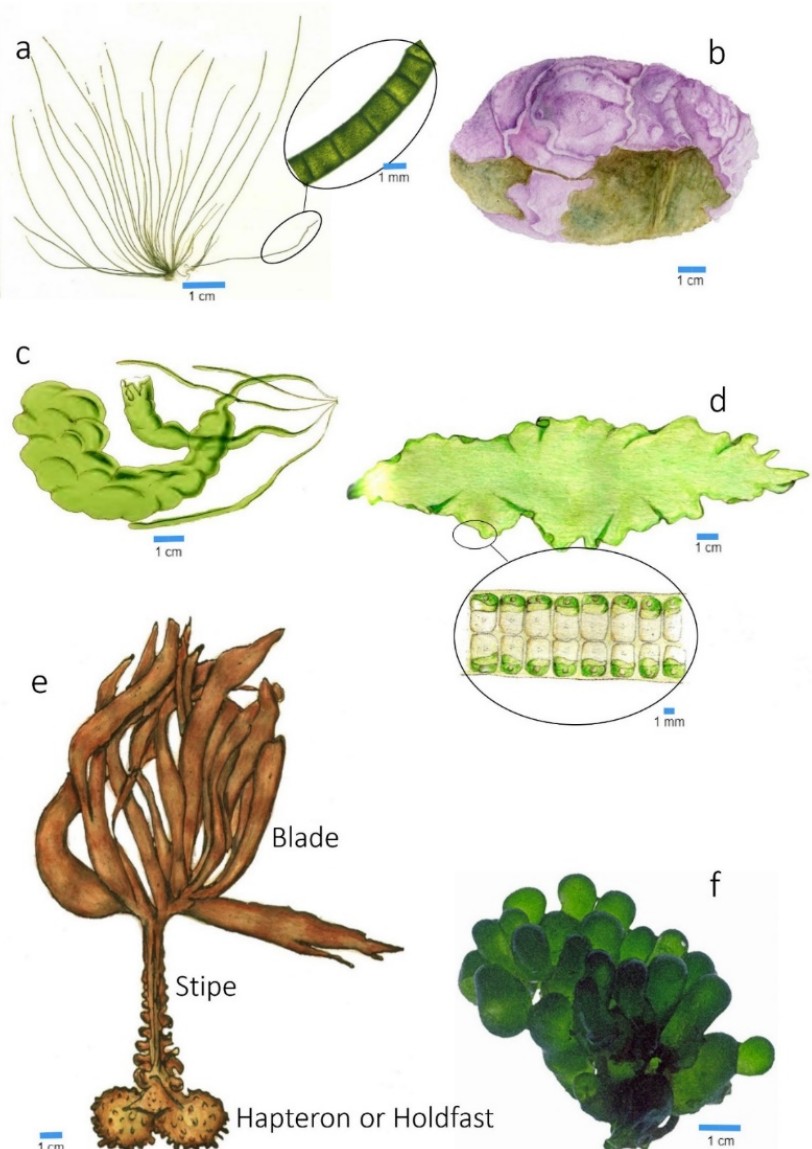

**Figure 4.** Different morphological types of macroalgae thalli: (**a**) cylindrical; (**b**) encrusting; (**c**) hollow tubes; (**d**) blades; (**e**) kelp thallus type; (**f**) vesicular.

Some algae are filamentous; in them the frond may be reduced to a filament in which the cells meet one after the other, and the filaments may be straight or branched.

The consistency and texture to the touch of the thalli are very diverse. They can be cartilaginous, leathery or coriaceous, mucilaginous, spongy, etc., and some thalli, called calcareous (because their cells are impregnated with calcium carbonate), have a rock-hard consistency.

While some thalli have cylindrical axes, others are flattened, and others form hollow tubes. Some thalli form monostromatic or polistromatic blades or sheets (with one or more layers of cells, respectively), being thin, more or less thick, or even coriaceous; orbicular or elongated; divided or not; and lobed or deeply divided (laciniated blades, ribbons, straps or belts). These features can be traversed by "nerves" or "veins".

### 5.1.1. Filamentous Thalli

The frond can be reduced to a filament, in which the cells are placed, linearly, one after the other (see Figure 4a). The filaments can be simple or branched (in the form of a bush). The ramifications can be irregular, dichotomous, alternate, opposite, verticillate, or pectinate.

### 5.1.2. Massive Thalli

The massive stems usually have a compact structure, and some species may have a soft consistency and delicate texture. Massive thalli are subdivided into four distinct types:

1.  Cylindrical shafts, also called "strings", normally erect when immersed, sometimes appearing prostrate, that can branch out according to the abovementioned modalities;
2.  Cylindrical or flattened tubes (see Figure 4c), when the thalli are hollow, and their walls are composed of one or more layers of cells and the tube axes may present constrictions at regular intervals (articulated thalli);
3.  Sheets, monostromatic (one layer of cells) or polistromatic (two or more layers of cells) (see Figure 4d);
4.  Vesicular, when the thalli, usually polistromatic, have a globose shape (see Figure 4f).

The blades can be thin, more or less thick, or even leathery; orbicular or elongated; divided or not; and lobed or deeply divided (lacinated blades or belts). The blades (or straps) can be traversed by "ribs" (prominent, clearly visible formations), or by "veins" (rows of less prominent cells).

### 5.2. Thalli Growth

Thalli growth occurs in several ways:

1.  Apical, when the divisions are restricted to one or more cells in the apical zone of the thallus;
2.  Diffuse, when divisions occur in different parts of the stem;
3.  Marginal (laminar thalli), when only the most peripheral cells divide regularly;
4.  In kelp, growth occurs at the base of the meristem, where the blades and stipe meet.

## 6. Reproduction in Algae

Algae reproduction can be asexual, where fertilization does not occur, or sexual, characterized by the intervention of gamete fusion. Generally, a single individual is capable of reproducing both asexually and sexually [3,5].

### 6.1. Asexual Reproduction

In this type of reproduction, the formation of new individuals can be done through three distinct processes that invest in homogeneity; that is, the perpetuation of the genetic heritage of the "parent" is promoted (only mitosis intervenes in the exact duplication and division of material between two daughter cells):

1.  By thalli fragmentation, in which each piece originates a new thallus and reconstitutes a new individual;
2.  By means of propagules, that is, through a small cluster of cells, with the ability to attach to a substrate and functionally originate a new thallus;
3.  Through spores, which are cells formed inside sporocysts (specialized structures, originating from modified mother cells) that result from mitotic divisions of the nucleus of the mother cell. At the end of their differentiation, the spores are released to the outside of the sporocysts (into the aquatic environment) through an opening in their wall. Depending on the species, the spores may be immobile (aplanospores, floating in the water column) or mobile (zoospores, moving with scourges, using whip movements). In both cases, the spores have the purpose of promoting the dispersion of the species, attaching themselves to a new and distant substrate, germinating and originating a new stem there. Some spores are true forms of resistance (akinetes or

hypnospores), being provided with a thick and impermeable wall that allows them to pass brief periods of dormancy, essential characteristics necessary for survival in adverse environmental conditions.

### 6.2. Sexual Reproduction

This mode of reproduction involves the production of specialized cells through meiosis, the gametes, whose sexual role is manifested by attraction and subsequent fusion with the gamete of the opposite sex. This cellular fusion, which gathers genetic assets stored in gametes of the opposite sex, with half of the chromosomes that typify the species (haploidy/n), is called fertilization or gamma (in which the number of chromosomes is restored, the diploidy/2n).

In certain algae, gametes of different sexes have a similar morphology, and reproduction is isogamous. In the other algae, the two categories of gametes are morphologically different, and reproduction is heterogamous. In both cases, after the fusion of the gametes, a cell called an "egg" or "zygote" (2n) originates.

In this reproduction process, two types of fertilization can be distinguished:

1. Planogamy, isogamous or heterogamous, in which both gametes are mobile, and fertilization occurs between these mobile cells (similar or not), giving rise to a swimmer pair (plano-zygotes), during the period in which their fusion in the zygote occurs;
2. Anisogamy or heterogamy, in which the male gamete meets the female gamete, usually larger and most often devoid of flagella, and merges with it. In red algae, a particular type of oogamy occurs, as it occurs between different gametes, both of which are not flagellated. In this case, fertilization occurs in situ, that is, in the place where the female gamete formed.

### 6.3. Formation of Gametes

Most gametes are formed inside a mother cell or gametocyst. Gametocysts producing male gametes (spermatozoa or antherozoids) are called spermatocysts, and gametocysts that originate female gametes (oospheres) are called oogonia.

All these types of reproduction and fertilization lead to algae being characterized by presenting different strategies to ensure the survival and continuity of the species, constituting their life cycles. These biological cycles are characterized by exhibiting a whole organized and serial process of events, which ends up being repeated in form and function in many different species. Up to six types of life cycle models can be observed in algae: four define the strategies observed in green and brown macroalgae and two, the most complex, in red algae.

### 6.4. The Reproductive Structures of Red Algae

Red algae (Rhodophyta) are a widespread group of uni- to multicellular aquatic photoautotrophic plants. They exhibit a broad range of morphologies, simple anatomy, and display a wide array of life cycles. About 98% of the species are marine, 2% are freshwater, and a few are rare terrestrial/sub-aerial representatives [4].

Red algae are true plants in the phylogenetic sense since they share a single common ancestor with the green lineage (green algae e and higher plants) [5]. However, the phylum Rhodophyta is easily distinguished from other groups of eukaryotic algae due to a number of features listed below [4,6–8]:

1. Total absence of centrioles and any flagellate phase;
2. Presence of chlorophylls a and d, and accessory pigments (light-harvest) called phycobilins (phycoerythrin and phycocyanin);
3. Plastids with unstacked thylakoids and no external endoplasmic reticulum;
4. Absence of parenchyma and presence of pit-connections between cells (i.e., incomplete cytokinesis);
5. Floridean starch as storage product.

Traditionally, red algae can be morphologically separated in three major groups:

1. A unicellular group with reproduction by binary cell division only;
2. A multicellular group where a carpogonial branch is absent or incipient (Bangiophyceae sensu lato);
3. A multicellular group with well-developed carpogonial branches (Florideophyceae).

Species from Florideophyceae present development of a specialized female filament called the carpogonial branch. The female gamete (carpogonium) is easily recognizable by the presence of the trichogyne, an elongated extension responsible for receiving the male gametes (spermatia) (Figure 5a). Germination occurs in situ of the zygote along with consequent formation of a group of spores (carpospores) or a parasite generation of the female gametophyte, which produces carpospores (carposporophyte) inside the cystocarp.

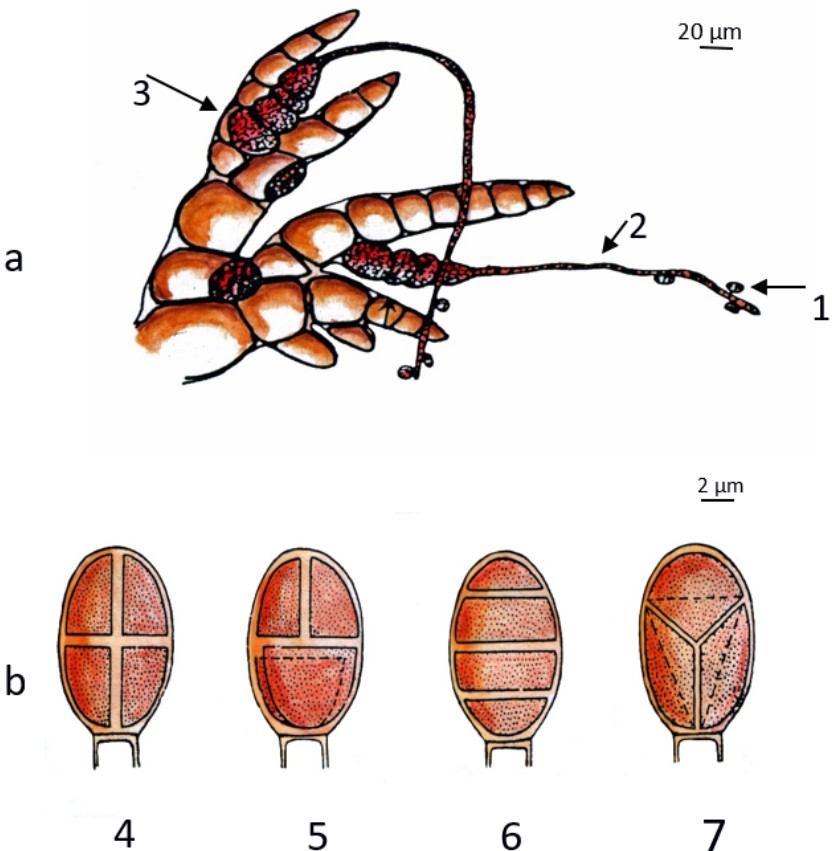

**Figure 5.** Reproductive cells of a red alga (phylum Rhodophyta): (**a**) the sperm cells (1—male gametes) are directed to the female organ in the carpogonial branch (2—trichogyne and 3—carpogonium) through the sea currents; (**b**) the different types of tetraspores (4—cruciate, 5—decussately cruciate, 6—zonate, 7—tetrahedral).

The cystocarp is composed of the carposporophyte plus all protective sterile haploid tissue of the female gametophyte encircling and interacting with it (pericarp). Carpospores develop into a second free-living phase called tetrasporophyte, which can be morphologically similar to (isomorphic alternation of generations) or different (heteromorphic alternation of phases) from the gametophytes. Tetrasporophyte thalli produce tetrasporangia by meiosis, which releases tetraspores (Figure 4b). This pattern of meiotic cell division in the tetrasporangium is stable in red algae and can be one of three types: cruciate, tetrahedral, or zonate (Figure 5b). When released, each tetraspore gives rise to either a male or a female haploid gametophyte [4,8].

In general, Florideophyceae present triphasic isomorphic or heteromorphic, diplohaplontic (haploid gametophyte, diploid carposporophyte and diploid tetrasporophyte), or diphasic diplohaplontic lifecycles [9–11].

### 7. Macroalgae Uses

Macroalgae offer a large number of possibilities for commercial use. Many tons of macroalgae are harvested annually around the world, with China and Japan being the countries that excel in their consumption. However, macroalgae are used in many countries for very different purposes, namely for food or for domestic animals (feed), agriculture, and medicine because of their therapeutic properties. The increasing worldwide use of macroalgae has led to them now being exploited in industry [2,12].

#### 7.1. Human Food and Animal Feed

Microalgae and macroalgae have been utilized by humans for hundreds of years as food, fodder, remedies, and fertilizers. Ancient records show that people collected macroalgae for food as long ago as 500 B.C. in China and one thousand years later in Europe. Microalgae such as *Arthrospira* (formerly *Spirulina*, Cyanobacteria) have a history of human consumption in Mexico and Africa. In the 14th century the Aztecs harvested *Arthrospira* from Lake Texcoco [13] (Farrar 1966) and used it to make a sort of dry cake called "Tecuitlatl", and very likely the use of this cyanobacterium as food in Chad dates back to the same period, or even earlier to the Kanem Empire (9th century A.D.). People migrating from countries such as China, Japan, and Korea, but also from Indonesia and Malaysia, where algae have always been used as food, have brought this custom with them, so that today there are many more countries all over the world where the consumption of algae is not unusual, including Europe [14].

Sea vegetables, as they are known, are included in the menus of many Eastern countries. On the contrary, consumption of macroalgae in Western countries was synonymous with penury. Macroalgae represent exactly the opposite of the foods that surround us today. They are a natural food that provides us with a high nutritional value and with low fat content. Although about 50% of the dry weight of macroalgae is carbohydrates, humans do not have the necessary enzymes to break these long molecules, so they are not absorbed by the digestive system, behaving like water soluble fibers. Macroalgae are extraordinary dietary supplements due to their high content of minerals, vitamins, and structural polysaccharides (fibers), which can facilitate intestinal transit and lower the cholesterol level in the blood [2].

Seaweeds also have a long history of use as livestock feed. They have a highly variable composition, depending on the species, time of collection and habitat, and on external conditions, such as water temperature, light intensity, and water nutrient concentration. The results available on the effect of seaweed supplementation on rumen fermentation are controversial, with reports of increased, decreased, or absence of effects. To be extensively used in ruminant feeding, seaweeds are requested in large quantities [15].

#### 7.2. Agriculture

The use of macroalgae in agriculture as fertilizers is one of the oldest traditional uses. Some Corallinaceae are used to correct the pH of acidic soils, while at the same time they increase the production of crops because they contribute certain elements such as magnesium, strontium, boron, and iron. Seaweed extracts contain several natural compounds (such as auxins, cytokinins, gibberellin, and others) that can improve plant development. There are many species of seaweeds, and their varied extracts have been tested to control different plant diseases and insects that cause damage to the plants. In general, there are different modes of action by which the seaweeds control plant diseases and insects. Plant pathology studies show positive results related to the effect of induced resistance in plant defense systems against pathogens by using seaweed extracts or their isolated compounds [15].

#### 7.3. Industry

The first industrial uses of macroalgae had the objective of producing calcium carbonate, soft drinks, glass, and soaps. However, macroalgae were also used for the extraction

of iodine as well as some dyes. Currently, the use of macroalgae is oriented toward the production of phycocolloids, producing organic compounds that form with the water colloidal systems, which are able to form consistent gelatins at room temperature [12,14].

The main compounds studied are fatty acids, pigments, phenols, and polysaccharides. Polysaccharides are the most exploited molecules and are already widely used in various industries [16]. For example, fucoidan is one of the most relevant polysaccharides produced by the Phaeophyceae class. The main bioactivities of fucoidan are anticoagulant, anti-tumor, anti-inflammatory, anti-thrombotic, or immunomodulatory, and these activities support the potential of this molecule to be explored by science and industry [17]. Regarding pigments, the carotenoid fucoxanthin, present in all brown algae, has activity antiangiogenic and protective effects against retinol deficiency [14].

### 7.4. Pharmaceuticals and Cosmetics

The phycocolloids have a preponderant role in the pharmaceutical industry due to their stabilizing properties, as thickeners, in the extraction of compounds with antiviral, antibacterial, or anti-tumoral action. Within the phycocolloids, it is important to emphasize the important role of agar as a medicinal use. Carrageenans are used in medicine as they can inhibit the development of the herpes virus and infection by the human papilloma virus. They are further useful in the treatment and washing of hair due to their ability to bond with keratin. Alginic acid is a complex polysaccharide extracted from brown algae (*Laminaria*, *Fucus*, and *Ascophyllum*). It is used in the medical and cosmetic industries because of its stability in wide ranges of pH and salinity, thus allowing fast healing and neutralization of certain heavy metals or in cases of intoxication by ingestion. Green macroalgae have been used as anti-worms. Red macroalgae are used as anticoagulants, anti-worms, and in the treatment of gastritis and diarrhea, while brown macroalgae are commonly used in menstrual disorders, hypertension, skin diseases, syphilis, and gastric ulcers and have an anticoagulant effect [12,14].

## 8. Global Production of Macroalgae

Global production of farmed aquatic algae, dominated by seaweeds, experienced relatively low growth in the most recent years, and even fell by 0.7% in 2018. This change was mainly caused by the slow growth in the output of tropical seaweed species and reduced production in Southeast Asia, while seaweed farming production of temperate and cold-water species was still on the rise [18].

Although coastal ponds for aquaculture, modern or traditional, are found in almost all regions in the world, they are far more concentrated in South, Southeast, and East Asia and Latin America for raising crustaceans, finfish, mollusks, and, to a lesser extent, seaweeds [18].

Over the last two decades, seaweed mariculture has taken place mainly within nine East and Southeast Asian countries and territories: from 98.9% in 2000 to 99.5% in 2018. Zanzibar (Tanzania) and Chile produced very small amounts (0.3 and 0.1%, respectively) of the world seaweed aquaculture production. All other countries in the world, combined, produced only 0.1% [19].

Most of this production happened in China, Indonesia, and other Asian countries (47.9%, 38.7%, and 12.8% of the worldwide production in 2016, respectively), mainly for human food and food additives. The total aquaculture production of seaweeds more than doubled in the last 20 years, and the total potential has been suggested to be 1000–100,000 million tons, but the main practice outside Asia is still to harvest natural stocks (García-Poza et al., 2020). The top 10 aquatic algae producers in 2018 were China, Indonesia, Philippines, Republic of Korea, Democratic People's Republic of Korea, Japan, and Malaysia [18,19].

In 2018, farmed seaweeds represented 97.1% by volume of the total of 32.4 million tons of wild-collected and cultivated aquatic algae combined. Seaweed farming is practiced in a relatively small number of countries, dominated by countries in East and Southeast Asia.

The world production of marine macroalgae, or seaweed, has more than tripled, up from 10.6 million tons in 2000 to 32.4 million tons in 2018 [18–20].

## 9. Conclusions

Macroalgae are aquatic photosynthetic organisms that do not constitute a defined taxonomic category, presenting organisms with different cellular organizations and a wide variety of morphologies and forms of growth that still generate intense discussions about their classification. Macroalgae, together with marine angiosperms, are essential primary producers for maintaining the marine ecosystem, with the microalgae that make up plankton being responsible for 40% to 50% of global primary production, while macroalgae stand out for their great economic importance [21].

All groups of macroalgae are currently attracting the attention of the scientific community due to the bioactive substances they produce. Several macroalgae extracts have exceptional properties with nutritional, nutraceutical, industrial, agricultural, pharmacological, and biomedical interest [12].

**Funding:** This work is financed by national funds through FCT-Foundation for Science and Technology, I.P., within the scope of the projects UIDB/04292/2020-MARE-Marine and Environmental Sciences Centre.

**Acknowledgments:** I thank Ignacio Bárbara (University of A Coruña, Galicia, Spain) providing an image of *Chondrus crispus* var. *filiformis*.

**Conflicts of Interest:** The author declares no conflict of interest.

**Entry Link on the Encyclopedia Platform:** https://encyclopedia.pub/7932.

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
