# Peer review of "Macroalgae"

_encyclopedia, doi:10.3390/encyclopedia1010017_

Round 1

Reviewer 1 Report

Review comments on “What are macroalgae and what distinguishes them from seagrasses”

General comment:

“What are macroalgae and what distinguishes them from seagrasses?” I think the addition of a question mark is important here.

From the above title it appears that the article will deal with a deeper comparison between macroalgae and seagrasses, but it is not the case. In fact, the article describes in great details on various aspects of macroalgae, with slight relevance with microalgae/cyanobacteria. Hence, the title can suitably be changed to “Macroalgae” only

Specific comments:

Ln10: Mention what is chlorophyll and what it does? From common people perspective.

Ln93: Provide an appropriate reference for this statement.

Table1: Check for typographical error for the carotenoids name. also in table legend, correct “pigmentary” to “pigment”

1.2. Pigment Composition and Classification: In this section there is no description of cyanobacteria except in the Table. So, update the section with relevant information or delete “Cyanobacterial information” from the table.

Figure 4 legend: Italicise genus and species name. It is not clear why used “versus”, it is obvious from their morphology how similar or dissimilar between the two species. So, I would recommend, label the photomicrographs corresponding to their species name.

Figure5: It will be very helpful if scales can be provided to understand the morphological variants of each of these forms of thalli.

Lines174-176: It is better to delete the part of cyanobacterial reproduction, as it is not relevant to the topic nor described in detail.

1.5.3. Industry: This section can further be improved with possible biotechnological uses of Fucoidan, Fucoxanthin, and other value-added molecules.

Also, add a concluding paragraph at the end of this article.

Author Response

Reviewer 1

Author: I am very grateful for the comments and suggestions made, as this way the manuscript will be effectively improved

General comment:

“What are macroalgae and what distinguishes them from seagrasses?” I think the addition of a question mark is important here.

From the above title it appears that the article will deal with a deeper comparison between macroalgae and seagrasses, but it is not the case. In fact, the article describes in great details on various aspects of macroalgae, with slight relevance with microalgae/cyanobacteria. Hence, the title can suitably be changed to “Macroalgae” only

Author: I agree with the comment, so the title will simply be "Macroalgae"

Specific comments:

Ln10: Mention what is chlorophyll and what it does? From common people perspective.

Author: a simplified definition has been added

Ln93: Provide an appropriate reference for this statement.

Author: reference added

Table1: Check for typographical error for the carotenoids name. also in table legend, correct “pigmentary” to “pigment”

Author: corrections made as indicated

1.2. Pigment Composition and Classification: In this section there is no description of cyanobacteria except in the Table. So, update the section with relevant information or delete “Cyanobacterial information” from the table.

Author: information updated with the pigments present in Cyanobacteria.

Figure 4 legend: Italicise genus and species name. It is not clear why used “versus”, it is obvious from their morphology how similar or dissimilar between the two species. So, I would recommend, label the photomicrographs corresponding to their species name.

Author: label information corrected

Figure5: It will be very helpful if scales can be provided to understand the morphological variants of each of these forms of thalli.

Author: scales have been added in Figure 5

Lines174-176: It is better to delete the part of cyanobacterial reproduction, as it is not relevant to the topic nor described in detail.

Author: Cyanobacteria information has been removed

1.5.3. Industry: This section can further be improved with possible biotechnological uses of Fucoidan, Fucoxanthin, and other value-added molecules.

Author: have been added to this item possible biotechnological uses of several value-added molecules extracted from seaweeds.

Also, add a concluding paragraph at the end of this article.

Author: brief conclusions have been added

Reviewer 2 Report

The author has constructed a broad overview of macroalgae that touches on many topics.  I am sympathetic to the fact that when approaching an article of this type it is difficult to strike a balance between breadth and depth.  Overall I feel that this article does a decent job covering many of the points but delves too deeply in some areas (e.g. red algae sexual reproduction) while leaving others nearly unmentioned (e.g. ecological importance of macroalgae). 

I feel that this manuscript would benefit from extensive revision so that there is better balance between the topics.  In my mind, an encyclopedia article such as this should give readers a ‘high altitude’ view of macroalgae and provide citations within each section that can be pursued for more detailed study.  My suggestions for main topic headings are:

  • Taxonomy
  • Morphology and physiology
  • Reproduction
  • Ecology
  • Human value
  • Status of wild populations and aquaculture production

Detailed comments:

Line 23-30:  very long sentence with multiple parentheticals, consider breaking up to multiple

Line 37: anabolism à anabolic

Line 40: I think you can remove the types of microscopes.

Line 43-48:  Long sentence that forms its own paragraph.  Break up

Line 53:  thallus needs definition, also, are thalli always highly complex?

Line 58: They à Macroalgae

Line 61:  Hedgehogs?

Line 62:  hanging ending parenthesis ‘)’, also, do only herbivores find refuge from predators?

Line 64:  how does an algae improve its defense?  Adaptation is not a willful process…

Lines 65-67:  these statements require citation

Figure 2:  This figure is a composite of several others and does not depict what the text or the caption state.

Line 80: respective to what?

Line 81: consensual à the general consensus

Line 91:  ‘some algae’?  why not all? Or a defined subset?  As written it comes off as a random sampling. 

Line 95-98: this sentence needs re-structuring

Line 97:  or thallus?  As used previously?

Line 101: most hit?  What does this mean?

Figure 3:  do these  illustrations need citation?

Line 121:  This sentence needs restructuring.  The correlation between thallus size, structure and texture is not inherent or clearly related.

Line 125: mucilaginous (remind of mucilage) <- not sure how that is descriptive. In this sentence I think all descriptors in parentheses can be removed, e.g.: ‘leathery… (remind of leather)’.  This doesn’t do much to improve familiarity.

Line 134-136:  repetitive

Lines 137-169:  this list needs formatting to make it clear.

Lines 146-147:  how big does a diameter need to be to be ‘considerable’?

Figure 5:  text is cut off, illustrations uncited?

Lines 172-173: the reproductive strategies of macroalgae are diverse but, arguably, not general.

Section 1.4.2:  I don’t think an explanation of the evolutionary merits of sexual reproduction are necessary here.  To me, this section should only contain information relevant to macroalgae.

Section 1.4.4.  This is a detailed description of reproduction in one type of macroalgae, at the expense of discussing others.  For this article I would keep it simple and broad, covering the basics of all three main classes of macroalgae.

Lines 296-307:  Detailed historic records of algae consumption is a bit exhaustive for this scope.

Line 308:  What is the best known use?

Lines 295-325:  This whole section can be abbreviated and outlined more broadly. Hit the key points: 1) historical use, 2) Asia dominates consumption 3) high nutritional value, 4) compounds also useful, 5) animals benefit too.

Section 1.5.3:  by contrast, this section is perhaps overly brief.  There are a lot of industrial uses for seaweed products.  A few more sentences here would be enlightening.

Author Response

Reviewer 2

Author: I am very grateful for the comments and suggestions made, as this way the manuscript will be effectively improved

The author has constructed a broad overview of macroalgae that touches on many topics.  I am sympathetic to the fact that when approaching an article of this type it is difficult to strike a balance between breadth and depth.  Overall I feel that this article does a decent job covering many of the points but delves too deeply in some areas (e.g. red algae sexual reproduction) while leaving others nearly unmentioned (e.g. ecological importance of macroalgae). 

Author: I agree with the comment, but as the referee mentioned very well, it is difficult to write about this topic (Macroalgae) and to be able to address all possible associated themes

I feel that this manuscript would benefit from extensive revision so that there is better balance between the topics.  In my mind, an encyclopedia article such as this should give readers a ‘high altitude’ view of macroalgae and provide citations within each section that can be pursued for more detailed study.  My suggestions for main topic headings are:

  • Taxonomy
  • Morphology and physiology
  • Reproduction
  • Ecology
  • Human value
  • Status of wild populations and aquaculture production

Author: the suggestions for main topic headings, made by the referee, are essentially those that make up the manuscript in the submitted version, with the last paragraph (in the revised version) addressing a chapter on macroalgae and their sustainability (see the last bibliographic reference added)

Detailed comments:

Line 23-30:  very long sentence with multiple parentheticals, consider breaking up to multiple

Author: done

Line 37: anabolism à anabolic

Author: done

Line 40: I think you can remove the types of microscopes.

Author: done

Line 43-48:  Long sentence that forms its own paragraph.  Break up

Author: done

Line 53:  thallus needs definition,

Author: Since it is the caption in figure 1, it would not be appropriate to add a broad definition of thalli

also, are thalli always highly complex?

Author: phrase has been adjusted

Line 58: They à Macroalgae

Author: corrected

Line 61:  Hedgehogs?

Author: corrected for “sea urchins”

Line 62:  hanging ending parenthesis ‘)’,

Author: corrected

also, do only herbivores find refuge from predators?

Author: I think it is not necessary to lengthen the sentence further

Line 64:  how does an algae improve its defense?  Adaptation is not a willful process…

Author: I think it is not necessary to lengthen the sentence further

Lines 65-67:  these statements require citation

Author: done

Figure 2:  This figure is a composite of several others and does not depict what the text or the caption state.

Author: I think the figure represents the role of algae in a marine habitat

Line 80: respective to what?

Author: I think that this question only arises due to the positioning of Figure 2, so I changed its position in order to be closer to the text area that is referred to

Line 81: consensual à the general consensos

Author: The sentence has been simplified

Line 91:  ‘some algae’?  why not all? Or a defined subset?  As written it comes off as a random sampling. 

Author: the title of Table 1 is correct, since the algae to which this manuscript refers are mainly macroalgae

Line 95-98: this sentence needs re-structuring

Author: done

Line 97:  or thallus?  As used previously?

Author: from my point of view they are not synonymous

Line 101: most hit?  What does this mean?

Author: hit by waves (corrected)

Figure 3:  do these  illustrations need citation?

Author: Figure 3 is cited in line 85

Line 121:  This sentence needs restructuring.  The correlation between thallus size, structure and texture is not inherent or clearly related.

Author: setence removed

Line 125: mucilaginous (remind of mucilage) <- not sure how that is descriptive. In this sentence I think all descriptors in parentheses can be removed, e.g.: ‘leathery… (remind of leather)’.  This doesn’t do much to improve familiarity.

Author: corrected

Line 134-136:  repetitive

Author: removed

Lines 137-169:  this list needs formatting to make it clear.

Author: corrected

Lines 146-147:  how big does a diameter need to be to be ‘considerable’?

Author: I do not think it is necessary to specify a value!

Figure 5:  text is cut off, illustrations uncited?

Author: corrected

Lines 172-173: the reproductive strategies of macroalgae are diverse but, arguably, not general.

Author: corrected

Section 1.4.2:  I don’t think an explanation of the evolutionary merits of sexual reproduction are necessary here.  To me, this section should only contain information relevant to macroalgae.

Author: remoded in part

Section 1.4.4.  This is a detailed description of reproduction in one type of macroalgae, at the expense of discussing others.  For this article I would keep it simple and broad, covering the basics of all three main classes of macroalgae.

Author: The reproduction of algae of the Florideophyceae class is the most complete and complex of all algae, so it deserves special attention

Lines 296-307:  Detailed historic records of algae consumption is a bit exhaustive for this scope.

Author: the historical and current use of algae in food is the main use, hence the emphasis given to this use

Line 308:  What is the best known use?

Author: Setence remoded

Lines 295-325:  This whole section can be abbreviated and outlined more broadly. Hit the key points: 1) historical use, 2) Asia dominates consumption 3) high nutritional value, 4) compounds also useful, 5) animals benefit too.

Author: corrected

Section 1.5.3:  by contrast, this section is perhaps overly brief.  There are a lot of industrial uses for seaweed products.  A few more sentences here would be enlightening.

Author: corrected

Round 2

Reviewer 1 Report

Review comments on “Macroalgae”

General comment:

Move the “Figure 2 legend” below the figure as that of other figures.

Table1: Change “mixoxanthophyll” to “myxoxanthophyll”. Not sure if there is any carotenoid found in cyanobacteria called “equine” please check if this a typographical error. Write in foot note what (+) and (-) means in the column for “Phycobilins”.

Author Response

Reviewer 1

Review comments on “Macroalgae”

General comment:

Move the “Figure 2 legend” below the figure as that of other figures.

Author: Figure 2 was eliminated

Table1: Change “mixoxanthophyll” to “myxoxanthophyll”.

Author: Corrected

Not sure if there is any carotenoid found in cyanobacteria called “equine” please check if this a typographical error.

Author: Word removed

Write in foot note what (+) and (-) means in the column for “Phycobilins”.

Author: The meaning of (+) and (-) has been added

Reviewer 2 Report

The Author has revised this manuscript to address several key areas that needed clarification but I feel that some of the broader structural issues I had with the initial submission remain unaddressed in the current form.  As I stated in my previous comments, I appreciate the difficulty in writing an encyclopedia article that is simultaneously broad and deep and, frankly, don’t envy the author in this task. 

In the last round of comments I offered a suggested list of topics to address in this MS, and the author replied that this is exactly what this article contains.  I disagree.  This article does a good job covering taxonomy, physiology, reproduction and commercial/industrial uses of macroalgae.  In my opinion, the topics of ecology, sustainability of populations, and aquaculture are lacking, and warrant mentioning in an encyclopedic article. 

A general note, I think a brief ‘table of contents’ would be good to have after the definition to direct readers to sections that they may be specifically interested in.  This is more of a formatting suggestion for the publisher. 

I have a few specific comments to this draft that I have listed below and, to be clear, these are my suggestions, not demands, and I am just providing my input as a reviewer.

Line 50: constituting -> constitute

Line 53:  ‘famous’ sardines?  I think that the use of these kinds of adjectives detract from the scientific gravity of the work.

Line 60:  Again, thalli is undefined.  The word does appear in figure one but it is not defined there and terminology like this needs to be described in the text, not captions.

Line 61: Again, the thalli of some macroalgae are complex, but others are not, this sentence needs to allow for this variation.  For example: “ Macroalgae, or seaweeds, are macroscopic marine algae, which can reach several meters in length… with highly organized tissues and varying levels of complexity”

Line 62:  Starting the sentence with ‘they’ is very ambiguous and should be changed (mentioned last round).  One suggestion is something like: “As primary producers, macroalgae are at…”

Line 73: benthic should be removed.  Herbivorous fish may be benthic, demersal or semi-pelagic.

Figure 2: I still have issues with this figure and caption.  The figure is an artificial mash up of a kelp forest, a Mola Mola (a pelagic predator) and some sort of grouper (a demersal predator).  Overall, I feel it does little to accentuate the concepts in the text (even contradicts them?) and does not depict the ecology of macroalgal forests.

Line 97:  The use of ‘respectively’ in this context requires the assignment of an ordered list of items to a second one.  For example:  “Ulva and sugar kelp are green and brown algae, respectively”.  In this case, the assigned item immediately follows the categorical one (e.g. ‘Plantae (green and red algae)’) so ‘respectively’ is relating no two such ordered lists.

Line 114: “most thalli” -> “The thalli of most macroalgal species are erect”.

Line 120:  “most hit by waves coastal areas” -> “in coastal regions with high wave exposure”

Figure 4:  suggested caption: “Red macroalgae Chondrus crispus (left) and Chondrus crispus var. filiformis (right).”

Line 143:  With the move in text chunk to above, the reintroduction of the ‘fixation part’ is abrupt and more or less out of place.

Line 187:  Again, I don’t think the term ‘considerable’ is a scientifically precise term to use here.  If the author feels that no minimum diameter is a necessary threshold for his classification then why is it required that they reach a ‘considerable’ diameter?

Line 209:  Where does thallus growth from the meristem (i.e. in Saccharina spp.) fit in these categories? 

Line 219:  I guess I may be confused with what this sentence states.  The way I read ‘sexual and asexual generations coexist in the same individual’ is that one organisms is composed of cells derived from both reproductive strategies.  What I think the author means is that a single individual is capable of reproducing both asexually and sexually.  If it is the former, that is a new concept to me that would need a citation I believe.  In either scenario, this needs to be worded precisely so that readers understand what the author is stating. 

Line 280:  This section starts off with a heavy exposition on red algae that is unrelated to reproductive structures.  As stated previously, it is also unbalanced detail for this group relative to the green and brown algaes.  Even if the increased reproductive complexity of this phylum warrants further explanation, I find it hard to argue that it deserves to deprive explanations of reproduction in other species (e.g. Ulva).

Lines 347-377:  please see my comment to the previous draft, which the author says were addressed but remain unchanged.

Author Response

Reviewer 2

The Author has revised this manuscript to address several key areas that needed clarification but I feel that some of the broader structural issues I had with the initial submission remain unaddressed in the current form.  As I stated in my previous comments, I appreciate the difficulty in writing an encyclopedia article that is simultaneously broad and deep and, frankly, don’t envy the author in this task. 

Author: I am very grateful for the reviewer's comments and opinions, but in fact most of the suggested changes go in the opposite direction to my original idea about this manuscript, so in general (except for spontaneous situations), I don't see how to be able to include them in this review.

In the last round of comments I offered a suggested list of topics to address in this MS, and the author replied that this is exactly what this article contains.  I disagree.  This article does a good job covering taxonomy, physiology, reproduction and commercial/industrial uses of macroalgae.  In my opinion, the topics of ecology, sustainability of populations, and aquaculture are lacking, and warrant mentioning in an encyclopedic article. 

Author: Regarding the reviewer's opinion, that the topics of ecology, population sustainability and aquaculture are missing and deserve to be mentioned in an encyclopedic article, my opinion that these topics should be in another separate manuscript, and not in this one, as the inclusion of the ecology, sustainability and aquaculture would make this manuscript gigantic.

A general note, I think a brief ‘table of contents’ would be good to have after the definition to direct readers to sections that they may be specifically interested in.  This is more of a formatting suggestion for the publisher. 

Author: An index may be included, but it must be the publisher's (or editor) decision and not mine

I have a few specific comments to this draft that I have listed below and, to be clear, these are my suggestions, not demands, and I am just providing my input as a reviewer.

Author: I appreciate all possible suggestions that improve the manuscript.

Line 50: constituting -> constitute

Author: Done

Line 53:  ‘famous’ sardines?  I think that the use of these kinds of adjectives detract from the scientific gravity of the work.

Author: The word "famous" has been removed

Line 60:  Again, thalli is undefined.  The word does appear in figure one but it is not defined there and terminology like this needs to be described in the text, not captions.

Author: The following sentence has been added at the beginning of 1.3 section: Thallus (plural "thalli"), is the plant body of algae, fungi, lichens, and some liverworts. A plant body that is not differentiated into stem and leaves and lacks true roots and a vascular system.

Line 61: Again, the thalli of some macroalgae are complex, but others are not, this sentence needs to allow for this variation.  For example: “ Macroalgae, or seaweeds, are macroscopic marine algae, which can reach several meters in length… with highly organized tissues and varying levels of complexity”

Author: Phrase removed

Line 62:  Starting the sentence with ‘they’ is very ambiguous and should be changed (mentioned last round).  One suggestion is something like: “As primary producers, macroalgae are at…”

Author: "As primary producers" was added, as suggested

Line 73: benthic should be removed.  Herbivorous fish may be benthic, demersal or semi-pelagic.

Author: Removed

Figure 2: I still have issues with this figure and caption.  The figure is an artificial mash up of a kelp forest, a Mola Mola (a pelagic predator) and some sort of grouper (a demersal predator).  Overall, I feel it does little to accentuate the concepts in the text (even contradicts them?) and does not depict the ecology of macroalgal forests.

Author: Figure 2 removed

Line 97:  The use of ‘respectively’ in this context requires the assignment of an ordered list of items to a second one.  For example:  “Ulva and sugar kelp are green and brown algae, respectively”.  In this case, the assigned item immediately follows the categorical one (e.g. ‘Plantae (green and red algae)’) so ‘respectively’ is relating no two such ordered lists.

Author: word removed

Line 114: “most thalli” -> “The thalli of most macroalgal species are erect”.

Author: Corrected

Line 120:  “most hit by waves coastal areas” -> “in coastal regions with high wave exposure”

Author: Corrected

Figure 4:  suggested caption: “Red macroalgae Chondrus crispus (left) and Chondrus crispus var. filiformis (right).”

Author: Corrected

Line 143:  With the move in text chunk to above, the reintroduction of the ‘fixation part’ is abrupt and more or less out of place.

Author: I readjusted the position of the figures in order to be closer to the quotation in the text.

Line 187:  Again, I don’t think the term ‘considerable’ is a scientifically precise term to use here.  If the author feels that no minimum diameter is a necessary threshold for his classification then why is it required that they reach a ‘considerable’ diameter?

Author: Removed from the text

Line 209:  Where does thallus growth from the meristem (i.e. in Saccharina spp.) fit in these categories? 

Author: the following sentence was added “4. In kelps, growth occurs at the base of the meristem, where the blades and stipe meet”.

Line 219:  I guess I may be confused with what this sentence states.  The way I read ‘sexual and asexual generations coexist in the same individual’ is that one organisms is composed of cells derived from both reproductive strategies.  What I think the author means is that a single individual is capable of reproducing both asexually and sexually.  If it is the former, that is a new concept to me that would need a citation I believe.  In either scenario, this needs to be worded precisely so that readers understand what the author is stating. 

Author: Corrected with the last option and added two citations numbers.

Line 280:  This section starts off with a heavy exposition on red algae that is unrelated to reproductive structures.  As stated previously, it is also unbalanced detail for this group relative to the green and brown algaes.  Even if the increased reproductive complexity of this phylum warrants further explanation, I find it hard to argue that it deserves to deprive explanations of reproduction in other species (e.g. Ulva).

Author: The reproduction of macroalgae and their respective life cycles will be the subject of another manuscript, different from this one, since it is already too long

Lines 347-377:  please see my comment to the previous draft, which the author says were addressed but remain unchanged.

Author: In the previous version some more sentences were added, as suggested. Due to the fact of having eliminated one of the figures, all the rest had their numbers changed, as well as their citation in the text.

Round 3

Reviewer 2 Report

The most recent version of this manuscript has accommodated most of my concerns, thank you for your effort.